# Hypoxia/HIF Modulates Immune Responses

**DOI:** 10.3390/biomedicines9030260

**Published:** 2021-03-05

**Authors:** Yuling Chen, Timo Gaber

**Affiliations:** 1Charité—Universitätsmedizin Berlin, Corporate Ember of Freie Universität Berlin, Humboldt-Universität zu Berlin, and Berlin Institute of Health, Department of Rheumatology and Clinical Immunology, Charitéplatz 1, 10117 Berlin, Germany; yuling.chen@charite.de; 2German Rheumatism Research Centre (DRFZ) Berlin, a Leibniz Institute, Charitéplatz 1, 10117 Berlin, Germany

**Keywords:** hypoxia, HIF, T cells, B cells, monocytes, macrophages, neutrophils, ILC, oxygen

## Abstract

Oxygen availability varies throughout the human body in health and disease. Under physiological conditions, oxygen availability drops from the lungs over the blood stream towards the different tissues into the cells and the mitochondrial cavities leading to physiological low oxygen conditions or physiological hypoxia in all organs including primary lymphoid organs. Moreover, immune cells travel throughout the body searching for damaged cells and foreign antigens facing a variety of oxygen levels. Consequently, physiological hypoxia impacts immune cell function finally controlling innate and adaptive immune response mainly by transcriptional regulation via hypoxia-inducible factors (HIFs). Under pathophysiological conditions such as found in inflammation, injury, infection, ischemia and cancer, severe hypoxia can alter immune cells leading to dysfunctional immune response finally leading to tissue damage, cancer progression and autoimmunity. Here we summarize the effects of physiological and pathophysiological hypoxia on innate and adaptive immune activity, we provide an overview on the control of immune response by cellular hypoxia-induced pathways with focus on the role of HIFs and discuss the opportunity to target hypoxia-sensitive pathways for the treatment of cancer and autoimmunity.

## 1. Introduction

Immune cells and proper immune response require focal sites of immune cell development, maturation, activation, tolerance, and longevity also defined as immunological niches bearing a certain microenvironment to maintain immune homeostasis [1]. These organs and tissues include the bone marrow, placenta, intestinal mucosa, renal medulla, secondary lymphoid organs, and the thymus [2,3]. In tissue pathology, sites of high immunological activity lead to inflammation and as a result tissue dysfunction bearing certain pathological microenvironment features. These pathological sites include infected, inflamed, and ischemic tissues and tumors [4,5,6,7]. Of note, sites of immune activity with distinct microenvironmental entities can broadly range between a state of immune homeostasis and a state of immune pathology. Under certain conditions of severe and disorganized immune activity, inflammation can perpetuate as a result of immune dysfunction leading to autoimmunity or culminates into inflammation-driven tumor development [8]. Microenvironmental conditions at sites of physiological and pathological immune activity play a key role in the development of effective immune response and pathological immune dysfunction by modulation of immune cell function. Understanding the impact of the microenvironment in sites of immune activity and adaptation mechanisms in immune cell reprogramming may yield into new therapeutic treatment strategies against a dysfunctional immune response as found in autoimmunity and cancer. At sites of immune activity under physiological and pathophysiological circumstances, immune cells become highly metabolically active and activate bystander cells and surrounding tissue. As a result, microenvironmental features rapidly change by increasing the amount and number of humoral factors, metabolites and a decrease in oxygen leading to a state of hypoxia—a condition where cellular oxygen demand exceeds the oxygen supply [9,10]. Constant supply of oxygen is a prerequisite for the energy homeostasis of respiring cells. Oxygen plays a vital role in all eukaryotes, being the terminal electron acceptor of the mitochondrial electron transport chain, which finally feeds the proton gradient for the generation of ATP via oxidative phosphorylation. If the constant supply with oxygen does not anymore meet the requirements of cells, hypoxic conditions will be established and, if sustained, these conditions will ultimately result in cell death. Hypoxia arises in a variety of immunological situations under physiological and pathophysiological immune activity [10,11].

## 2. Physiological Hypoxia Influences Immunity

Fundamental principle of the vasculature is to supply all organs, tissues and cells with oxygen and nutrients according to their needs and to dispose of refuse (carbon dioxide and metabolic products) establishing a balance between supply and consumption which is unique for the respective organ, tissue and cell. With regard to oxygen, its availability to the cells in the human body depends on various factors, such as (i) oxygen uptake, (ii) the transport capacity of the blood, (iii) the transport of the oxygen carrier, i.e., vascularization, and finally (iv) cell respiration itself.

Even under physiological conditions oxygen partial pressure (pO_2_) varies throughout the human body (Figure 1) [12,13,14,15]. Arterial blood owns an average oxygen partial pressure of ~80–100 mmHg which corresponds to an oxygen air-content (O_2_ air-content) at sea level of 10–12.5%. The extreme values are 100 mmHg in the pulmonary veins and 40 mmHg in the pulmonary arteries. The tissue oxygen partial pressure varies depending on the tissue anatomy and function in the range of 30–50 mmHg (~3–6% O_2_ air-content) dropping to a cellular range of 9.9–19 mmHg (~1–2% O_2_ air-content) and further to a mitochondrial pO_2_ of <9.9 mmHg (~1% O_2_ air-content) [13]. Consequently, current standardized cell culture conditions are oriented towards of atmospheric pO_2_ with oxygen concentrations 2–5 times higher than physiologically relevant, which are ignoring in vivo situation [12,13].

Low pO_2_ have been detected in various compartments of healthy and inflamed tissues as well as in tumors, often as a characteristic of tissue architecture, vascularization and microenvironment [1]. Tissues and cells vary in their (i) circulatory distance from lung oxygenation (ii) density, functionality and relative proximity of/to their capillary network (iii) oxygen consuming microenvironmental, (iv) the rate of oxygen consumption within the cells and thus in pO_2_ leading to distinct thresholds and susceptibilities to hypoxia [12,14,15]. However, at a cellular level, hypoxia and hypoxic responses generally occur at a pO_2_ ~7–10 mmHg (~1% O_2_ air-content) [17].

Although most tissues of the body are provided with a level of oxygen that exceeds the basal metabolic requirements, in some tissues, the pO_2_ is comparatively low, which results in regions of “physiological hypoxia” [1]. Such regions can occur in the intestinal outermost mucosal surface where a controlled oxygen gradient establishes as a result of anatomical features such as juxtapositioning of the mucosal surface to the anoxic gut lumen and the functional countercurrent oxygen exchange system in the intestinal villi [18]. In kidney, oxygen gradients are necessary for organ function which is to maximize the concentration of urine by counter-current exchange of oxygen in the renal medulla [13,19]. Moreover, oxygen gradients are important for the synthesis of erythropoietin (EPO) in kidney and liver [20]. In the developmental process of the placenta and the fetus, physiological hypoxia can be observed in several regions due to constant outgrowing of the existing local blood supply [21]. If blood supply is limited due to the lack of vasculature such as found in the eye’s retina but also in the outer layer of the skin, the epidermis, ‘physiological hypoxia’ has been demonstrated to be well established [21,22]. Moreover, the major organs of the immune system, including bone marrow (pO_2_ < 10 mmHg) [23,24], thymus (pO_2_ < 10 mmHg) [24,25,26], spleen (pO_2_ ~ 4–34 mmHg) [26,27], and lymph nodes (pO_2_ < 4–46 mmHg) [28], exhibit regions of immune activity with locally significantly lower pO_2_ than surrounding tissues and even lower than inhaled air. These hypoxic regions are of functional importance because they impact immunity by providing a niche for hematopoietic stem cells (HSCs) in the bone marrow, where hypoxia maintains the self-renewal capacity of HSCs favors a slow turnover of HSCs and sustains survival by promoting their quiescence [29,30,31,32] or an environment for the antigen challenging of B cells in germinal centers (GCs), where hypoxia increase glycolytic metabolism supporting the generation and expansion of antigen-specific GC B cells and the production of high-affinity immunoglobulin G (IgG) antibodies [33,34].

## 3. Pathophysiological Hypoxia Shapes Immune Response

Hypoxia also exists in pathophysiological states, which are more severe and confused as compared to physiological hypoxia [35] (Figure 2). Solid tumor microenvironment is one of the well-known typical pathophysiological immunological hypoxia caused by an imbalance between oxygen supply and oxygen demand [36]. The rapidly proliferating tumor cells are outgrowing from the vascular network, which limits the diffusion of oxygen into the intratumor microenvironment leading to hypoxia. In the hypoxic microenvironment of the tumor induces proangiogenic factors, such as vascular endothelial growth factor (VEGF), and promotes tumor vascularization and growth. However, the tumor’s blood vessels are usually irregularly structured and poorly functional, and also tend to form clots and local edema, aggravating local hypoxia. Moreover, the tumor induced neovasculature has gaps between endothelial cells resulting in that tumor cells leak into bloodstream and disseminate [37,38,39]. The hypoxic milieu recruits myeloid-derived suppressor cells (MDSCs) to the primary tumor site by activating the transcription of chemokine ligand in cancer cells [40] and promoting ectonucleoside triphosphate diphosphohydrolase 2 (ENTPD2/CD39L1) [41]. MDSCs play a key role in tumor immunosuppression by inhibition of anti-tumor T cell effector function. Usually MDSCs inhibit antigen-specific CD8+ T cells in lymphoid organs thereby reducing collateral damage and controlling effector function, but MDSCs at the tumor site preferentially differentiate into tumor associated macrophages (TAMs) facilitated by the hypoxic tumor microenvironment inhibiting not only antigen-specific but also nonspecific T cell activity [42]. Furthermore, the hypoxic microenvironment increases the expression programmed death ligand 1 (PD-L1) on MDSCs, which is when blocked resulting in enhanced MDSC-mediated T cell activation [43] evidencing hypoxia-mediated suppression of anti-tumor T cell effector function supporting tumor development [44,45,46].

Apart from the hypoxic tumor sites, hypoxic areas may appear as a consequence of infection with pathogenic bacteria, viruses, fungi, and protozoa [47,48,49,50]. Many factors contribute to the establishment of an hypoxic environment including an increased oxygen consumption by inflamed resident cells, infiltrating immune cells and pathogens as well as a decreased oxygen supply caused by the combination of vascular pathology and microthrombosis [51]. In this scenario, the hypoxic microenvironment protects the host by decreasing host cell death and reducing pathogenicity of invaders, while deleterious effects such as increases in antibiotic resistance and bacterial invasion make hypoxia a double-edged sword [52,53,54].

However, sites of inflammation undergo significant shifts in metabolic activity leading to O_2_ deficiency, which is defined as “inflammatory hypoxia” [35,55]. The reasons for this kind of hypoxia include the increase in oxygen consumption by infiltration and transmigration of immune cells such as monocytes and polymorphonuclear neutrophils (PMN), by local T and B cell proliferation, by activation of oxygenase, such as oxidases, monooxygenases and dioxygenases, and the immunometabolic switch in effector cells itself. The influence of inflammatory hypoxia on the severity of inflammation is particularly tissue-specific and depends on the composition and distribution of the involved cell types, the local microenvironment, the duration and severity of hypoxia [56].The induction of hypoxia by tissue infiltrating PMNs for instance in the intestinal epithelia ameliorates colitis [56], while in the lung enhances the severity of lung injury [57].

Blocking the blood supplying vessels by thrombus, embolus, or other blockages and followed by the subsequent restoration of perfusion and concomitant reoxygenation leads to ischemia and reperfusion injuring the demanding tissue [58]. Ischemia and reperfusion often occur in small capillaries of cerebral, coronary, or peripheral arteries [59] and is one of the leading causes of morbidity and mortality. Imbalance of oxygen supply and demand during ischemia and reperfusion leads to tissue hypoxia and immune cell attraction, which trigger inflammation and result in the tissue damage in ischemic disease [60,61].

## 4. Hypoxia-Inducible Factors (HIF’s): Structure, Function, and Regulation

Hypoxia itself influences cellular behavior in different ways ranging from reduction of oxidative phosphorylation to induction of glycolysis leading to lactate production and acidification of the cellular microenvironment. Cells respond to these changes by inducing a certain specific transcriptional program induced by a variety of transcription factors.

### 4.1. HIF’s—“Master Regulators” of Cellular Response to Hypoxia

The key regulator of hypoxia-induced transcriptional changes in all metazoan—also termed as “master regulator” of the adaptive transcriptional response to hypoxia—is the heterodimeric hypoxia inducible factor family of transcription factors (HIF’s) [62,63]. Three members namely HIF-1, HIF-2 and HIF-3 belong to this family, bearing distinct functional activities and specificities. The best characterized transcriptional factor today of the cellular hypoxic response is the HIF-1 which binds to hypoxia response elements (HREs) in the promoters of a variety of genes to activate a transcriptional program that is directed towards adaptation to hypoxia by inducing a metabolic shift towards glycolysis, promoting oxygen transport, cell migration, the re-establishment of oxygen supply by inducing angiogenesis and vasculogenesis [64,65]. Although several of these targets are conserved in different cell types, HIF-1 also confers a variety of cell-type specific transcriptional responses towards a hypoxic environment [65] (Figure 3).

The HIFs are mainly composed of two basic helix-loop-helix (bHLH) proteins which dimerize via a PAS (Per, AHR/ARNT, Sim) domain, as defined by sequence homology to the Drosophila Per and Sim proteins and the mammalian aryl hydrocarbon receptor (AHR) and aryl hydrocarbon receptor nuclear translocator (ARNT) proteins [66,67] to form the transcription factor HIF (Figure 3A). The heterodimeric HIF consists of an oxygen-dependent alpha-subunit (stabilized under hypoxia) and a constitutively expressed beta-subunit (HIF-1β or ARNT). Three α subunits have been described so far: HIF-1α, HIF-2α, and HIF-3α. While HIF-1α and HIF-2α are well characterized, relatively little is known about HIF-3α [68,69,70,71,72]. HIF-1α regulation itself ranges from transcriptional to translational and posttranslational level [73].

In the presence of sufficient oxygen, when cellular oxygen supply exceeds demand, HIF-α becomes hydroxylated on prolyl residues (402 and/or 564 in HIF-1α, 405 and/or 531 in HIF-2α) by dioxygenases termed prolyl hydroxylase domain proteins (PHD1-3) (Figure 3B) [74]. Hydroxylation facilitates the interaction with the tumor suppressor protein von Hippel–Lindau (pVHL), which recruits an E3-ubiquiine-ligase complex tagging HIF by polyubiquitination for proteasomal degradation via the 26S proteasome [75,76,77]. Furthermore, a second dioxygenase termed the factor-inhibiting HIF (FIH) hydroxylates HIF on an asparagine residue (Asn-803 in HIF-1α) and inhibits the transcriptional activation of HIF by blocking the interaction with the transcriptional coactivators p300 and CREB (cAMP response element binding)-binding protein (CBP) [68,74,78,79].

Under hypoxic conditions where the oxygen demand exceeds the supply, and/or due to a redox imbalance, dioxygenase activity of the PHDs and FIH is attenuated and HIF-α becomes stabilized and de-repressed, respectively. Once stabilized, HIF-α translocates into the nucleus, dimerizes with the constitutively expressed ARNT, recruits transcriptional coactivators (p300/CBP) and binds to HREs activating the cell-specific adaptation program towards hypoxia [10,68,74]. 

Two transcriptionally active HIFs, HIF-1 and HIF-2 confer the transcriptional activation and cellular adaptation under hypoxic conditions being distinguished by their α-subunits (HIF-1α and HIF-2α). Although both bind the HIF-1β, they demonstrate a distinct tissue expression profile with overlapping but also distinct functions [63,79,80]. In immunity, HIF-1 seems to be expressed in virtually all innate and adaptive immune cell populations including myeloid lineages including monocytes, macrophages and neutrophils [80,81], dendritic cells [82], T cells and B cells and NK cells and innate lymphoid cells (ILCs) (reviewed in [83,84,85]) while HIF-2 has been demonstrated in certain immune cells such as in tumor-associated macrophages, in vitro expanded CD8+ T cells and in response to cytokines and hypoxia (reviewed in [85,86]).

Dynamics of HIF-1α/HIF-2α stabilization vary with regard to acute, chronic and cyclic or intermittent hypoxia [87]. Notably, cyclic hypoxia is characterized by periodic exposure to cycles of hypoxia and reoxygenation (H–R cycles). Interestingly, several in vitro studies revealed that cyclic hypoxia increased HIF-1α levels to a greater extent than chronic hypoxia by e.g., ROS-mediated stabilization of HIF-1α [88,89], while chronic hypoxia showed a greater extent of HIF-2α stabilization as compared to cyclic hypoxia [90,91].

In addition to the O_2_-dependent HIF degradation pathway, there are some other pathways that activate posttranslational degradation of HIF-α in an O_2_-independent way. For example, the tumor suppressor p53 binds HIF-1α and does recruit mouse double minute 2 homolog (Mdm2) mediated ubiquitination and proteasomal degradation of HIF-1α in an oxygen-independent manner [92]. Another example is given by IF-1α degradation via an oxygen-independent E3 ubiquitin ligase using heat shock protein 90 (Hsp90) inhibitors [93]. Furthermore, HIF stability has been demonstrated to be susceptible to high partial pressure of CO_2_ (hypercapnia) which reduces intracellular pH and promotes lysosomal degradation of HIF-α subunits [94]. The lysosomal degradation pathway of HIF-α subunits has been also identified in the regulation of cell cycle progression by the physical and functional interaction of cyclin dependent kinases with HIF-α subunits [95,96].

Apart from the O_2_-independent posttranslational and translation regulation of HIF’s, HIF-α can be induced by a variety of inflammatory stimuli such as bacterial products (e.g., lipopolysaccharide or LPS), TNF-α and IL-1β which can lead to the activation of pathways successively engaging phosphatidylinositol 3-kinase (PI3K), Protein kinase B (PKB), also known as Akt and the highly conserved Ser/Thr kinase mammalian target of rapamycin (mTOR) [97,98,99] and/or the mitogen activated protein kinases (MAPK) pathways. These pathways promote cap-dependent translation of HIF-1α mRNA [100] or induce transcription factor nuclear factor kappa-light-chain-enhancer of activated B cells (NFκB)-dependent upregulation of HIF expression and activity [101,102,103,104,105]. Beside NFκB- and mTOR-dependent regulation of HIF, other pathways directly or indirectly control HIF and the HIF-pathway such as reactive oxygen and nitrogen species (ROS and RNS) [106,107,108]. Increased mitochondria-derived ROS levels during hypoxia mainly promote the stabilization of HIF-α subunits by inactivation of PHDs [109,110]. Full enzymatic activity of PHDs requires substrates and cofactors, such as ascorbate and ferrous iron [Fe (II)], which could be depleted by increased ROS [110]. ROS mediates the oxidation of cysteine residues, which form the catalytic site of PHD2, leading to generation of inactivated PHD2 homodimerization [109]. ROS also increases HIF-1α by other mechanism including the activation of the HIF-1α promoter via a functional NFκB-site [111] and inhibition of the hydroxylase domain of FIH [112]. Nitric oxide (NO) plays a paradoxical role in modulating HIF stability. High concentrations of NO (>1 µM) stabilizes HIF-1α under both hypoxic and normoxic conditions, but lower concentration of NO (<400 nM) facilitates HIF-1α degradation under hypoxic conditions [113]. This could be explained by the fact that under hypoxic conditions oxygen is redistributed toward oxygen-sensing hydroxylases such as PHDs and FIH due to NO-mediated inhibition of oxygen consumption by the mitochondrial cytochrome c oxidase (complex IV) [114]. Furthermore, NO might also induce directly the expression of PHDs [115,116].

### 4.2. Cellular Response to Hypoxia Beyond HIF’s

Cellular responses initiated by hypoxia governs a variety of cellular changes including signaling cascades, transcriptional, translational, and posttranslational changes as well as changes in metabolite pattern. Apart from HIF’s, several other transcription factors are involved in the cellular hypoxic response [117]. Most important with a certain role in intermittent hypoxia or cycling hypoxia (hypoxia reoxygenation cycles) is the activation of the transcription factor nuclear factor kappa-light-chain-enhancer of activated B cells (NFκB) [118], inducing transcription of many genes in the response to inflammatory stimuli as mentioned above including Hif-1α itself [118,119].

Moreover, the hypoxic response also includes the posttranscriptional regulation of gene expression via hypoxic-elicited microRNAs (hypoxamiR) including the master hypoxamiR miR-210. The latter is capable to simultaneously regulate the expression of multiple target genes in order to fine-tune the adaptive response of cells to hypoxia. HypoxamiR do this by regulation of HIF itself and, thus, cellular adaptation to low oxygen availability including the metabolic reprogramming, DNA repair, apoptosis, and angiogenesis [120,121,122,123,124].

Hypoxia-induced changes on transcriptional levels are relatively well understood, but hypoxia-induced responses on translation are not yet well demonstrated. The latter includes the downregulation of oxidative stress-induced misfolded and unfolded proteins caused by endoplasmic reticulum (ER) stress [125] and upregulation of essential proteins for cell survival [38].

Hypoxia-induced repression of protein translation occurs on different levels:Inhibition of cap-dependent translation initiation is mediated by the unfolded protein response (UPR) sensor, PERK (protein kinase RNA-like ER kinase), which phosphorylates the eukaryotic initiation factor 2 (eIF2α) attenuating protein synthesis [126,127].Hypoxia-induced repression of protein synthesis during peptide elongation and termination is mediated via suppression of eukaryotic elongation factors (eEFs) [128,129,130,131].Hypoxia-mediated accumulation of abnormal proteins at the ER activates IRE1 (inositol-requiring protein 1) and ATF6 (activating transcription factor 6) to deal with ER stress by facilitating protein-folding and maintaining cell survival [38,132].

Additionally, short-time exposure to hypoxia (30min) inhibits mTOR, which is the main regulator of cap-dependent translation and its effectors: eukaryotic translation initiation factor 4E (eIF4E)-binding proteins (4E-BPs), p70 S6 kinase, rpS6, and eukaryotic translation initiation factor 4G (eIF4G). This effect is independent of ATP levels, ATP:ADP ratios, and HIF1α [133]. However, in chronic hypoxia mTOR complex 1 (mTORC1) is repressed by AMPK (AMP-activated protein kinase) and REDD1 (regulated in development and DNA damage response 1). Hypoxia-induced upregulation of REDD1 and activation of the energy-sensor AMPK promotes activation of the TSC1/2 complex and then represses mTORC1 signaling and/or leads to eEF2 inhibition [134,135,136]. Thus, hypoxia-mediated inhibition of mTORC1 suppresses mRNA translation through many pathways, such as (i) activation of 4E-BPs, or (ii) inactivation of eEF2 [131,137].

To maintain specific gene expression and their translation under hypoxia, such as the expression of HIF target genes, mechanisms have evolved to circumvent hypoxia-induced global translational inhibition. These mechanisms mainly include the use of internal ribosome entry site (IRES)-mediated translation [138,139], hypoxic eIF4F(H)-mediated translation [140], the RNA-binding protein RBM4 and the cap-binding eIF4E2-mediated translation [141,142], as well as signal recognition particle (SRP)-dependent mRNA localization-mediated translation [143].

### 4.3. Mitochondrial Response to Hypoxia

Hypoxia reduces the cells oxygen consumption by inhibiting mitochondrial function and structural integrity including the electron flux via the electron transport chain (ETC) to the terminal electron acceptor oxygen (Figure 4).

Initially, hypoxia promotes the conversion of pyruvate into lactate but reduces the entry of pyruvate into the TCA cycle (tricarboxylic acid cycle) and its conversion to acetyl–CoA [144,145,146]. The reduction of acetyl–CoA which usually fuels the TCA cycle now leads to a reduction of energy intermediates and metabolites such as nicotinamide adenine dinucleotide (NAD) and flavin adenine dinucleotide (FAD) resulting in NADH and FADH_2_ (oxidized by the ETC to finally generate ATP) as well as aspartate (consumed in the process of nucleotide synthesis and essential for cell proliferation) [147]. Besides limiting supply of NADH and FADH_2_ for ETC, hypoxia also modifies ECT by regulation the activity of mitochondrial complexes [38].

Mitochondrial complex I and III of ETC are major sites of ROS production [145]. Although decreased ETC flux might be supposed to attenuate hypoxic ROS generation in general [148,149,150]. Paradoxically, hypoxia increase ROS production, which in turn stabilizing HIF-1α [151]. Recent studies demonstrated that the mitochondrial Na^+^/Ca2^+^ exchanger (NCLX) is activated under hypoxia leading to a mitochondrial increase in Na^+^. Accumulation of Na^+^ reduces the mitochondrial inner membrane fluidity and thus, the mobility of free ubiquinone. The immobile free ubiquinone leads to an increase of semiquinone at the Qo site of the mitochondrial complex III increasing the production of ROS under hypoxia [152]. Finally, hypoxia promotes mitochondrial fission mediated by mitochondrial protein fission 1 (FIS1) and activates mitophagy [153,154,155]. However, the mechanisms underlying how hypoxia modulates mitochondrial fission and mitophagy are not yet understood.

To compensate for reducing energy supply from mitochondrial metabolism, hypoxic cells also undergo a variety of metabolic changes to adapt to changes in the environment, such as enhancing glycolysis, glutaminolysis, fatty acid synthesis and serine synthesis pathway activity; decreasing pentose phosphate pathway activity, gluconeogenesis, nucleotide synthesis, fatty acids β- oxidation, lipid desaturation. Most of these mechanisms of metabolic reprogramming involve the regulation and modulation by the HIF- and the AMPK-pathway [38]. Additionally, hypoxic energy starvation inhibits mTOR independent of HIF and AMPK which increases over time [133,135,136]. As already mentioned, other hypoxia-responsive pathways include endoplasmic reticulum (ER) stress and NFκB pathways [102,119,156,157].

## 5. Hypoxia-Inducible Factors in the Regulation of Immune Response

Immune cells face a variety of states of oxygen availability from luxury oxygen supply in the vasculature to hypoxic sites of immunological activity and severe hypoxic pathological sites of inflammation where they have to remain functionally active. Using gain and loss of function of individual HIF subunits and their regulators (PHDs) in distinct immune cell subtypes using transgenic mice, have gained deep insights into the impact of HIF’s in immune response (reviewed in [158]).

### 5.1. HIFs in Myeloid Cell Function

Neutrophil granulocytes belong to the first line of defense which—recruited by macrophages—migrate from the circulation to sites of infection and inflammation into a pathophysiological hypoxic environment where their main function is to kill pathogens via the production of ROS (thereby reducing oxygen thus aggravating hypoxia), the formation of neutrophil extracellular DNA traps, the release of antimicrobial substances and proinflammatory cytokines like TNF-α, IL-1β, interferons for the recruitment and activation of other immune cells [159,160]. Furthermore, neutrophilic respiratory burst can reduce immune activity by contributing to “inflammatory hypoxia” and a decrease in neutrophil invasion which results in an effective inflammatory resolution in a murine model of intestinal inflammation [56]. However, the role of HIF-1α in neutrophils during this scenario remains unclear [161] (Table 1).

Furthermore, it has been demonstrated that activation of the HIF-1 pathway in neutrophils increases their survival and effector function (enhancement of β2 integrin expression and migration, production of antimicrobial peptides, the formation of DNA traps, phagocytosis and oxidative burst) and induces metabolic reprogramming by inducing the Warburg effect for the generation of ATP [9,80,81,162,163,178]. Moreover, a similar importance of neutrophilic HIF-2 for survival, inflammation and tissue injury has been demonstrated by Thompson et al. [164]. Interestingly, in patients with inflammatory diseases HIF-2 is elevated in neutrophils. Thus, activation of HIF-1 and HIF-2 can be considered to enhance a proinflammatory phenotype in neutrophils.

In contrast to neutrophils, granula-rich mast cells, basophils and eosinophils play a central role in Th2-mediated allergic reactions and in the protection against parasite and microbial infections. Again, HIF-1α stabilization in these cells remains important for survival and function as demonstrated in mice and humans, providing energy by HIF-1α mediated metabolic re-programming towards a glycolytic and controlling effector function by the synthesis of IL-8, TNF-α, VEGF, and IL-4 following TLR and/or IgE activation, the formation of DNA traps and chemotactic capacity [165,166,167,168,169].

The main actors of the innate immune response are the proinflammatory classically activated macrophages (M1-type) which belong to the first line of defense against foreign pathogens e.g., bacterial infection. M1 macrophages release pro-inflammatory cytokines such as TNF-α, IL-1β and IL-6 and produce large amounts of ROS by NADPH oxidase (further reducing oxygen availability promoting hypoxia) and nitric oxide (NO) via inducible nitric oxide synthase (iNOS) [179]. Conversely, alternatively activated macrophages (M2-type) facilitate wound healing and support tissue repair and regeneration by anti-inflammatory and proangiogenic activity producing arginase-1 (Arg-1), VEGF and IL10 [179,180,181,182]. M1- and M2-type macrophages differ also metabolically [183]. While short-lived M1 macrophages primarily rely on glycolysis and own reduced flux through the electron transport abolishing oxidative phosphorylation [184], M2 macrophages reduce their glycolytic flux [185] and make use of fatty acid oxidation and oxidative phosphorylation [186].

In M1 macrophages, the TCA cycle halts and intermediates such as citrate and succinate accumulate [184]. Succinate is oxidized by succinate dehydrogenase, thereby elevating mitochondrial ROS which contributes to the stabilization and increased activity of HIF-1α [182,185,186]. Secondly, LPS-induced succinate stabilizes HIF-1α and enhances IL-1β production [187]. Interestingly, both, HIF-1 and HIF-2, contribute to an increase in bactericidal activity, myeloid cell infiltration and an increase in acute inflammatory responses of M1 macrophages [81,163,173,188], but it seems that mainly HIF-1 is essential for the regulation of glycolytic capacity in myeloid cells feeding the cellular ATP pool which promotes myeloid cell survival, aggregation, motility, invasiveness, and bacterial killing [81], while HIF-2 modulates macrophage migration without altering intracellular ATP levels [188]. Moreover, HIF-1 has been demonstrated to be induced by Th1 cytokines in M1 macrophage polarization, whereas HIF-2 is induced by Th2 cytokines during an M2 response [171]. In contrast, using a model of sterile tissue damage Gondin et. al. demonstrated that M2 polarization is independent of either HIF isoform [170] assuming that only HIF-1 mediated metabolic reprogramming contributes to M1 polarization while the role of HIF-2 remains to be elucidated.

Dendritic cells (DCs) facilitate antigen-specific T-cell activation after recognition of invading pathogens or endogenous danger signals by their pattern recognition receptors (e.g., TLRs) [189]. Upon activation DCs increase glycolysis activate iNOS and NO production thereby inhibiting the electron flux through the electron transport chain [190]. Thus it is not surprising that HIF-1 has been shown to promote cell survival of mature DCs, DC migration, differentiation and proper DC-mediated cytotoxic T cell activation (involving IFN-α and -β production) and induction of regulatory T cells (Tregs) while inhibiting survival of immature DCs [174,175,176,177]. We can conclude from these data that proper innate immune response needs functional HIF’s at least to mediate metabolic re-programming.

### 5.2. HIFs in Adaptive Immunity

The adaptive immune response against cancer and infections is controlled and orchestrated by the T cell compartment. T cells originate from lymphoid progenitors in the bone marrow and mature in the cortex of the thymus, which is known to be hypoxic [25] into various subpopulations of helper CD4^+^ T cells and CD8^+^ cytotoxic T cells [191] (Table 2). Forced stabilization of HIF’s by deletion of VHL in thymocytes dramatically decreased thymic cellularity [192]. Additional deletion of HIF-1α restored thymocyte development suggesting a role of HIF-1α but not HIF-2α and the hypoxia pathway in lymphocyte development [192]. Conversely, forced stabilization of HIF’s does not affect neutrophil and monocyte development [81].However, these mature but naïve T cells are metabolic quiescent and rely on oxidative phosphorylation of glucose and lipids to maintain cellular homeostasis [193].

Upon antigen challenging, naive T cells undergo metabolic reprogramming towards aerobic glycolysis and glutamine catabolism and differentiate into different subpopulations of effector T cells thereby generating an immune memory [193,194]. Depending on their environmental cues, CD4^+^ T cells can differentiate into subpopulations such as Tregs, Th1, Th2, or Th17 cells, which have distinct metabolic patterns and immunological functions [193,194].

A prerequisite for effector T cell differentiation (such as Th1, Th2, or Th17 cells) is the upregulation of glycolysis whereas Tregs display increased lipid oxidation and oxidative phosphorylation [193]. Inhibition of glycolysis and/or glutaminolysis or forcing the utilization of fatty acids instead results in inhibition of effector T cell differentiation, T cell anergy or shifting the effector T cells towards regulatory T cell lineage [193,194].biomedicines-09-00260-t002_Table 2Table 2HIF-mediated effects in lymphoid cells.Cell TypeHIF Mediated EffectsRef.CD4^+^ T cells**HIF-1:**survival ↑,glycolysis ↑,Th17 differentiation and effector function↑,Th1 differentiation ↑, effector function ↓ (IFN-γ↓),Treg differentiation↓,Tr1 differentiation ↓[192,195,196,197,198,199,200,201,202]**HIF-2:**Treg differentiation ↓[203]CD8^+^ T cells**HIF-1:**survival ↑ glycolysis ↑effector function ↑ (anti-viral infection ↑, antitumor capacity ↑; IL-13 ↑)CD8^+^ T_C_2 cell differentiation ↑effector molecule↑[27,204,205,206,207]**HIF-2:**survival ↑effector function ↑ (IFN-γ and TNF-α ↑)[208]B cells**HIF-1:**survival ↑,glycolysis ↑,cell cycle ↑,effector function (IgG and IgM antibodies↓, IL-10 ↑),chemotherapeutic antitumor effect ↓[209,210,211]**HIF-2:**unknown
↑: induced or up-regulated, ↓: reduced or down-regulated.

HIF-1α has been demonstrated to play a crucial role in regulating development, survival, proliferation, and differentiation in murine and human T cells [192,195,199,200]. Activation of HIF-1 further promotes the metabolic reprogramming towards glycolysis thereby enhancing effector cell function. As one result, HIF-1α shifts the balance from Tregs to effector Th17 cells by (i) promotion of glycolysis [200] but also (ii) by binding of the Treg-specific transcription factor Foxp3 and its subsequent degradation favoring pro-inflammatory effector Th17 cells which drive inflammation and can lead to autoimmunity [199,200]. Moreover, HIF-1 itself is further upregulated by STAT3 and induces transcription of the gene encoding the retinoic acid-related orphan receptor γt (RORγt) [199]. Of note, RORγt is the key transcription factor that drives Th17 differentiation [212]. Finally, HIF-1 has been demonstrated to promote the longevity of Th17 cells by controlling Notch signaling and antiapoptotic gene expression which could be abrogated by HIF-1 inhibition using echinomycin treatment [213].

Apart from deleting VHL, HIF-1 mediated overexpression of miR-210 finally promotes autoimmunity (psoriasis-like inflammation) by inducing Th1 and Th17 cell differentiation [124]. Conversely, HIF-1α is not only enhancing miR-210, but also controlled in a negative feedback by miR-210 itself [123], an effect that is considered to be more important in Th1 cells. Here, HIF-1 negatively regulates Th1 function by reducing their capacity to produce IFN-γ [122]. Finally, the role of HIF-1 and miR-210 in Th17 and Th1 needs further investigations while studies on the role of HIF’s in Th2 cell response are still missing. Apart from HIF-1α which enhances Th17 differentiation and inhibits Treg cell differentiation, HIF-2α may also determine the fate of T cells. Although its expression was not sufficient in other Th subsets, HIF-2α was required for IL-9 expression in Th9 cells. Furthermore, miR-15b and miR-16 suppressed HIF-2α expression in Treg cells [203].

When focusing on Tregs, the impact of hypoxia and HIF’s is still less clear. Although HIF-1 has been reported to tip the balance of Th17/Tregs towards Th17 cells [199,200], some reports demonstrated that hypoxia enhances Treg abundance through HIF-1α–dependent regulation of FoxP3, thus, limiting inflammation [214,215]. Thus, hypoxia and HIF-1α in T cells constrain T-cell–mediated inflammation supporting earlier findings [216,217,218]. In this line, T cell-specific HIF-1α KO mice has been demonstrated to result in severe intestinal inflammation with an upregulation of Th1 and Th17 cells [219]. Conversely, Foxp3-restricted VHL deletion in mice augmented HIF-1α-induced glycolytic reprogramming and IFN-γ production and converted Treg cells into Th1-like effector T cells instead of Th17 cells which finally results in massive uncontrolled inflammation during dextran sulfate sodium (DSS)-induced colitis [220].

However, Foxp3-negative type I regulatory T cells (Tr1 cells), which own the capacity to attenuate TH17 cells are regulated by HIF-1α which controls the early metabolic reprogramming of human and murine Tr1 cells [196,197,198,201].

In the GC, activated extrafollicular CD4^+^ T cells and follicular T helper (Tfh) promote humoral immunity of B cells. HIF-1α depletion from CD4^+^ T cells reduces the frequencies of antigen-specific GC B cells, Tfh cells, Tfh/T follicular regulatory (Tfr) ratio, cytokine production (IFN-γ, IL-4),and overall antigen-specific antibody after immunization of sheep red blood cells (SRBC); while deficiency of HIF-1α and HIF-2α leading to further decrease in Tfh cells, Tfh/Tfr ratio, cytokine production and overall antigen-specific antibody. In contrast to this, after hapten-carrier immunization deficiency of HIF-1α has no effect on affinity maturation, while both deficiency of HIF-1α and HIF-2α lead to affinity maturation defects [202].

Effector CD8^+^ T cells or cytotoxic T lymphocytes (CTLs) are crucial mediators of cell-mediated immunity and play an essential role in the control of infections and cancer by the production and release of perforin and granzyme [221]. Hypoxia and HIF-1 favors the development of highly cytotoxic CD8^+^ T cells upon activation by facilitating metabolic reprogramming towards glucose uptake and glycolysis through the upregulation of numerous glycolytic enzymes while enhancing the expression of effector molecules such as granzymes and perforin [27,207]. Although both HIF-1α and HIF-2α are essential for controlling the survival, differentiation, and proliferation of human and murine CD8^+^ T cells [205], only HIF-1α, but not HIF-2α, was essential for the effector state in CD8^+^ T cells providing antitumor capacity [204]. Although rapid induction glycolysis, which is known to be facilitated by HIF-1, has been demonstrated to enhance effector memory CD8^+^ T cell function [206], limiting glycolytic flux and the expression of HIF-1 during activation of CD8^+^ T cells further enhanced the generation of long-lived memory CD8+ T cells and antitumor functionality [222], which was supported by the finding that commitment of CD8^+^ T cells to a highly glycolysis-driven metabolic program favored short-lived effector memory cells while negatively affecting long-lived memory populations [223]. A recent study on terminally differentiated tissue-resident T cells (CD69^+^CD103−D8^+^ T cells) of the human liver demonstrated that reduction of HIF-2α expression suppressed production of IFN-γ and TNF-α and increased TCR-induced cell death [208].

During chronic infection, the destructive capacity of CTLs is progressively dampened leading to ‘exhaustion’ which may be due induction of disease tolerance as a defense strategy [224,225]. Although stabilizing HIF’s by deletion of VHL lead to lethal CTL-mediated immunopathology during chronic infection bypasses T cell ‘exhaustion’ leading to an enhanced control of viral infection and tumor growth [226]. Moreover, hypoxia also enhance CD8^+^ type 2 cytotoxic T (T_C_2) -mediated inflammation and airway hyperresponsiveness via IL-4/HIF-1α-axis [227]. Finally, hypoxic CD8^+^T-cells demonstrate an increased capacity to persist, to proliferate, and to enhance antitumour efficacy upon a HIF-1α-dependent increase of metabolite S-2-hydroxyglutarate (S-2HG) production modifying epigenome [205].

In summary, these data demonstrate an important role for hypoxia and HIF-1-mediated signaling and metabolic reprogramming in the regulation of T cell function.

B cells play a crucial role in humoral immunity [228]. Upon their differentiation into plasmablasts and plasma cells B cells confer humoral immunity-by the secretion of antibodies protecting the host against an unlimited variety of pathogens. As effector cells, B cells are also involved in antigen presentation [228]. Moreover, B cells possess also immunomodulatory function by the secretion of anti-inflammatory cytokines such as IL-35, TGF-β and IL-10 [229,230,231,232,233,234]. Defects in B cell development leads to immune dysfunction resulting in immunodeficiency, allergy, autoimmunity and cancer [229,235].

Hypoxia and the HIF-1α pathway have been demonstrated to play a vital role in B cell activation, survival, proliferation, cell cycle arrest, development, effector and regulatory function [209,210,236,237]. In detail, B cell activation through B cell antigen receptor (BCR) induces upregulation of HIF-1α expression [209]. HIF-1α has been demonstrated to be essential for hypoxia-induced cell cycle arrest [237] and to be responsible for hypoxia-induced upregulation of two pore domain subfamily K member 5 potassium channels thereby augmenting Ca^2+^ signaling in B cells which is essential for number of cellular functions including proliferation, survival, and cytokine production [236]. In Hif1a/Rag2-deficient chimeric mice display B cell lineage defects including abnormalities in peritoneal B1 cells and high levels of IgG and IgM antibodies directed against dsDNA, a phenotype of autoimmunity [210]. Moreover, ablation of Hif1a, but not Hif2a in B cells impairs IL-10 production and the switch to glycolytic metabolism as well as CD1d^hi^CD5^+^ B cells expansion [209]. Ultimately, lack of Hif1a results in an exacerbated collagen-induced arthritis and experimental autoimmune encephalomyelitis [209]. HIF-1α along with p-STAT3 enhanced CD11b transcription in a DSS-induced colitis model. The latter commits regulatory function to B cells and is involved in protective activity in experimental inflammatory bowel disease (IBD) [238,239,240]. However, HIF-1α deficiency in B cells improves the chemotherapeutic antitumor effect by inhibiting CD19^+^ extracellular vesicles (EV) production [100]. CD19^+^ EVs hydrolyze ATP from tumor cells into adenosine which inhibits CD8 T cell antitumor response (see Table 2) [211].

In germinal centers (GCs), antigen challenging of B cells results in proliferation, expression of high-affinity antibodies, antibody class switching, and B cell memory. GC light zones have been demonstrated to be hypoxic [33]. GC hypoxia and HIF signaling increase glycolytic metabolism but reduced B cell proliferation, increased B cell death, impairs antibody class switching to IgG2c antibody isotype. Interestingly, constitutive activation and stabilization of HIF-1α induces B-cell proliferation, decreases antigen-specific GC B cells and impairs the generation of high-affinity IgG antibodies [33,34]. However, in hypoxic conditions the generation and maintenance of GC B cells has been demonstrated to rely on HIF-mediated increase of glycolysis and mitochondrial biogenesis for growth and proliferation [241]. In this line, blocking glucose uptake by deletion of glucose transporter 1 decreased B cell proliferation and impaired antibody production [242].

### 5.3. HIFs in Innate Lymphoid Immune Response

The innate lymphoid immune response is mediated by NK cells and so called innate lymphoid cells (ILCs). These tissue-resident immune cells do not express T cell receptors although they share similarities with CD4+ T cells and CTLs. ILCs are positioned at barrier surfaces to rapidly respond invading pathogens by local expansion to effectively initiate an immune response before adaptive immunity is engaged [243]. These areas are typically not well oxygenated at least when it comes to inflammation. Thus, it is essential that they own proper mechanisms to adapt to hypoxic conditions (Table 3).

ILCs can be classified into three different groups, namely group 1 ILCs (including ILC1s and NK cells), group 2 ILCs and group 3 ILCs (including lymphoid tissue inducer cells, natural cytotoxicity receptor positive and negative cells), based on their similarity to Th cell subsets in terms of their expression of key transcription factors and cytokine production [247]. ILCs protect against infectious pathogens such as helminths, viruses, intracellular parasite and protozoa, extracellular bacteria and fungi, and demonstrate functional plasticity depending on their inflammatory microenvironment [247].

Together with ILC1s, NK cells constitute group 1 ILCs, which protect against intracellular pathogens and are characterized by their capacity to produce interferon-γ and their functional dependence on the transcription factor T-bet [248]. NK cells commit cytotoxic activity releasing granzymes and perforins bearing anticancer and antiviral activity [249]. Like Th2 cells, group 2 ILCs are characterized by the need for Gata-3 and a Th2 cytokine profile to facilitate immune response against extracellular microorganisms [247].

Group 3 ILCs share similarities with Th17 cells by expressing RORγt and are implicated in tissue homeostasis, repair and inflammation [247]. All ILCs have been implicated when dysregulated leading to chronic inflammation and autoimmune diseases such as spondylarthritis (SpA), psoriasis and IBD [247].

Similar to CTLs, NK cells infiltrate hypoxic tumor areas in order to kill malignant cells. Loss of HIF-1 impaired tumor cell killing but still inhibited tumor growth by exacerbated VEGF-driven angiogenesis due to a reduced infiltration of NK cells that express angiostatic soluble VEGFR-1 [246]. However, hypoxia has also been demonstrated to impair NK cell cytotoxicity [250,251]. This seems surprising since tumor hypoxia inhibits NK cell function although these cells undergo metabolic reprogramming upon activation of effector function (IFN-γ production and granzyme B expression) from the quiescent state to glycolysis via the metabolic checkpoint kinase mTOR (in similarity to T cells) [252,253,254]. Interestingly metabolic reprogramming was only induced upon sustained activation but not after short-term priming [255]. Single cell RNA sequencing of mouse tumor-infiltrating NK cells identified Hif1a to be upregulated in the hypoxic microenvironment of solid tumors while conditional deletion of Hif1a^-/-^ in tumor-infiltrating NK cells activated the NFκB pathway, elevated expression of activation markers, effector molecule and increased INF-γ production in correlation with elevated oxygen consumption rate/extracellular acidification rate (OCR/ECAR) ratio finally reducing tumor growth [245].

In ILC2s deletion of VHL and stabilization of HIF’s caused a decrease in mature ILC2 numbers in peripheral nonlymphoid tissues without affecting early-stage bone marrow ILC2s, resulting in reduced type 2 immune responses. The decrease in mature ILC2 numbers results from HIF-1 mediated reduction of IL-33 receptor (ST2) expression [244].

However, how hypoxia and the HIF signaling pathway modulates function and metabolism of ILCs and NK cells in the setting of tumor biology and autoimmunity requires further research.

## 6. Extracellular Signals Regulating HIF’s in Immune Cells Apart from Hypoxia

Hypoxia and its impact on the stability of HIF and activity of HIF signaling fundamentally influences immune cell function in a cell-type specific manner. The temporal, spatial and dynamic nature of hypoxia differently impacts the immune cell function and response in a cell type specific manner. Under physiological conditions the temporal and spatial oxygen availability for cells and tissues remains constant. Although the degree of hypoxia varies between different tissues and from oxygen supplying capillaries to the cavities of mitochondrial lacuna, established oxygen gradients maintain stable and sustained allowing temporal and spatial limited fluctuations of the oxygen gradient [1].

Beside hypoxia, other extracellular microenvironmental factors (such as pathogens, cytokines, metabolic enzymes and metabolites) contribute to the induction and stabilization of HIF’s in immune cells (Figure 5). Under normoxia, Gram-positive and Gram-negative bacterial components induce HIF-1α expression and stabilization in mouse bone marrow-derived macrophages [163]. Treatment of macrophages with lipopolysaccharide (LPS), a major component of the outer membrane of Gram-negative bacteria, and associated activation of Toll-like receptors (TLRs) in macrophages, increases HIF-1α expression and leads to its stabilization by multiple pathways, such as PI3K pathway [256], p44/42 MAPK and NFκB pathway, but also by activation of pyruvate kinase M2 (PKM2) [257], and by accumulation of succinate after glutaminolysis [187,258]. Interestingly, the glycolytic enzyme pyruvate kinase M2 (PKM2) has been demonstrated to shape the cellular response to hypoxia thereby affecting the immune response. PKM2 controls the rate-limiting step of glycolysis and supports glycolysis and the biosynthetic processes [257,259,260]. Furthermore, in LPS-stimulated monocytes PKM2 has been demonstrated to translocate to the nucleus and to interact with HIF-1α thereby inducing the expression of inflammatory genes such as the expression of *IL1B*, *STAT3*, *HMGB1* (encoding IL-1β, signal transducer and activator of transcription, and high mobility group box 1) [257,259,260,261,262,263]. Furthermore, tumor necrosis factor-α (TNF-α) and interferon-γ (IFN-γ) are capable to increase HIF-1α mRNA expression in macrophages [264], while IL-4 and IL-13 increase HIF-2α mRNA abundance [171]. Moreover, oxidized low-density lipoprotein (oxLDL) induces HIF-1α accumulation by a ROS-dependent pathway in human macrophages [265]. Recent findings revealed that atheroma plaque homogenates increased human macrophages’ HIF-1α by forming liver-X-receptor (LXR)-HIF-1α-complexes on HIF-1α- and IL-1β-promoter-regions promoting inflammation in atherosclerosis [266]. Activation of mast cells with the calcium ionophore ionomycin enhanced HIF-1α gene and protein expression by activating calcineurin-dependent dephosphorylation of nuclear factor of activated T-cells (NFAT) thereby unleashing NFAT-dependent transcription [267]. Additionally, anti-IgE induces accumulation of HIF-1α protein in human basophils by activating extracellular regulated kinase (ERK) and p38 MAPK [169]. In mDC, neutralization of thymic stromal lymphopoietin (TSLP) and its receptor (TSLPR) during stimulation augments HIF-1α leading to an increased IL-1β expression [268].

Adaptive immune cells can also regulate HIF-1α in an oxygen-independent manner. Engagement of the T cell receptor (TCR) induces HIF-1α expression and accumulation in proinflammatory T helper 17 (Th17) and Th1 cells. Th17-polarizing conditions, presence of transforming growth factor-β (TGF-β) and IL-6, further enhance HIF-1α expression in a Stat3-dependent manner [199,200]. Moreover, TGF-β- and IL-23-induced HIF-1α to upregulate miR-210 expression, which promotes helper T (Th) 17 and Th1 cell differentiation [124]. However, CD4+T cell activation requires mitochondrial ROS [269]. The latter is well-known to induce and stabilize HIF-α [107,109,110]. Similar to PKM2 in monocytes, glycolytic enzyme activation may constitute another mechanism of HIF induction in T cell immune response. In brief, GAPDH binds to AU-rich 3′ UTR of several genes including *IFNG*, *IL2,* and *HIF1A* mRNA in glycolytic inactive naive and memory T cells. Upon T cell stimulation, glycolysis becomes activated occupying GAPDH thereby releasing *IFNG, IL2,* and *HIF1A* mRNA leading to effector cytokine production [74] and an elevated HIF-1α expression [75]. In CD8+ T cell, TCR-mediated signals can also mediate an increased abundance of both HIF-1α and HIF-2α which can be modulated by cytokines (e.g., IL-4, IL-2) [226]. HIF-1α mRNA and protein can be induced after LPS stimulation by NFκB pathway and BCR-stimulation via ERK–STAT3 pathway in B cells [209].

## 7. Summary and Outlook: How to Treat by Targeting HIF to Modulate Immunity

As outlined above, hypoxia and the HIF-response have a variety of implications on immune activity in physiological and pathophysiological context influencing the initiation and propagation of immune response and contributing to the development of immune dysfunction in autoimmunity and cancer. Thus, it is likely, that the hypoxia responsive pathways including HIF’s and PHDs could serve as promising therapeutic targets for pharmacological interventions.

Because of its significant impact on inflammation and immune-mediated inflammatory diseases (IMIDs) including autoimmune diseases and cancer, HIF-1α could serve as a promising therapeutic target. Unraveling the molecular mechanisms of the HIF-1α pathway and the evidence on the capacity of current treatment strategies to target this process may open novel therapeutic avenue to treat IMIDs. In fact, targeting HIF-1α in animal models of autoimmune diseases and cancer has yielded encouraging results and new pharmacological approaches. Consequently, a fast-developing domain of drug discovery has emerged targeting HIF-1α and/or HIF-2α to reduce or inhibit its transcriptional activity.

On the other hand, pharmacologic strategies to induce HIF stabilization have recently been tested in patients thereby setting the stage to use PHD inhibitors to treat patients suffering from diseases, such as chronic kidney disease and limb ischemia where the hyporesponsiveness of the HIF pathway has been observed [270]. Other potential clinical applications of HIF stabilizers include inflammatory hypoxia, such as occurs in the setting of inflammatory bowel disease, including ulcerative colitis and Crohn disease (CD). Here, activation of HIF induces intestinal barrier-protective genes (e.g., multidrug resistance gene-1, intestinal trefoil factor, CD73) and ensures mucosal integrity during colitis in vivo [271]. Dimethyloxalylglycine (DMOG), a pan-PHD inhibitor, has been demonstrated to be profoundly protective after intraperitoneal injection in a model of dextran-sodium sulfate colitis [272]. Targeted delivery of DMOG to the colon provides local protection not only resulting the same efficacy at a 40-fold lower dose as intraperitoneal injection, but also reduced systemic side effects in the treatment of colitis [273]. Pan-PHD inhibitors, FG-4497 [274] and TRC160334 [275], and HIF-1 isoform-predominant PHD inhibitor, such as AKB-4924 [56,276], also showed an attenuation of inflammation in murine colitis. Furthermore, regulation of IL-12p40 by HIF induced via AKB-4924 controls Th1/Th17 responses to prevent mucosal inflammation [277]. Importantly, HIF-2α, which can promote the inflammatory response and impair intestinal barrier integrity, constitutes a pathogenic mechanism in ulcerative colitis [278]. Due to the different role of HIF-1α and HIF-2α in ulcerative colitis, the balancing mechanisms of these subunits during therapeutic intervention to treat colitis needs to be further investigated [279]. Furthermore, in Crohn disease, another IBD, increased HIF-1α protein plays a major role in adherent-invasive *E. coli* (AIEC) induced inflammatory disorders of the gastrointestinal tract [280]. When targeting HIF pathways, HIF-1 seems to be a friend in the treatment of ulcerative colitis but a foe in other autoimmune-mediated pathogeneses such as Crohn disease, systemic lupus erythematosus, rheumatoid arthritis, and psoriasis [281].

Hypoxia and the induction of HIF within the tumor microenvironment has been demonstrated to promote cancer progression, metastasis, and therapy resistance by (i) reducing anti-tumor effector immune cells such as cytotoxic T cells, nature killer cells, and cytokines and/or (ii) increasing immunosuppressive cells such as Tregs and tumor-associated macrophages owing a M2 macrophage phenotype, the expression of immunosuppressive cytokines and inhibitory immune checkpoint molecules such as programmed death 1 (PD-1) receptor or Cytotoxic T-Lymphocyte-Associated protein 4 (CTLA-4) [282]. Many small molecules, which regulate each regulatory step of HIF-α-mediated gene expression in tumor cells from its (i) gene expression, (ii) protein translation, (iii) HIF-α stabilization, (iv) dimerization, and (v) DNA binding as well as (vi) HIF-α-mediated transcription of target genes have been developed. They are in majority targeting/inhibiting HIF-1α [270,283,284] instead of HIF-2α [285]. Apart from directly inhibiting HIF-1/2α, many molecules act on HIF-1/2-related signaling pathways, such as the PI3K/AKT/mTOR pathway, leading to indirect inhibition of HIF-1/2α [270,283,286]. However, HIF-1/2 regulation pathways are very complicated and contain interacting signaling networks. Thus, unspecific inhibition of HIF might result in unexpected side effects. Recently, selective HIF-2α allosteric inhibitors PT2385 and PT2977, which weaken heterodimerization with HIF-1β by selectively binding to HIF-2α-PAS-B domain [287], showed 66% and 80% of disease control rate and were well tolerated in phase I/II study of renal cell carcinomas (RCCs) [288,289,290]. A Phase III trial of monotherapy PT2977 in a population with previously treated RCC is planned and some clinical trials of selective HIF-2α inhibitors are underway (Table 4).

It is noteworthy that HIF-α actually improves effector function of many immune cells as explained before. Taking advantage of this knowledge, by selectively upregulating HIF-α in immune cells under conditions of a hypoxic tumor microenvironment may foster their tumor killing capacity; an effect which might provide a novel approach in the treatment of cancer.

Chronic kidney disease, ischemic disease and ulcerative colitis are beneficial from HIF stabilizer therapy, while cancers can be suppressed by HIF inhibitor therapy. In clinical application, the therapeutic window of HIF targeting should be exquisitely balanced. For example, the elderly tends to suffer from cancers and might be beneficial from HIF inhibitor, but this population is probably accompanied by ischemic heart disease which could be treated by HIF stabilizer. Pan-HIF inhibitor and stabilizer may have severe side effects, thereby their clinical application is limited. In these circumstances, local treatment and selectively HIF-1 or HIF-2 targeting medicine might provide therapeutic potential.

Although many clinical trials have confirmed that both HIF inhibitor and stabilizer can be used for clinical disease therapy, due to its widespread expression and wide impact on cell survival and function, the clinical benefits and side effects of drugs targeting HIF must be further carried out in specific disease context.

These and other findings reported and reviewed here suggest that targeting HIF-1α could be a useful strategy for autoimmune diseases therapies.

## Figures and Tables

**Figure 1 biomedicines-09-00260-f001:**
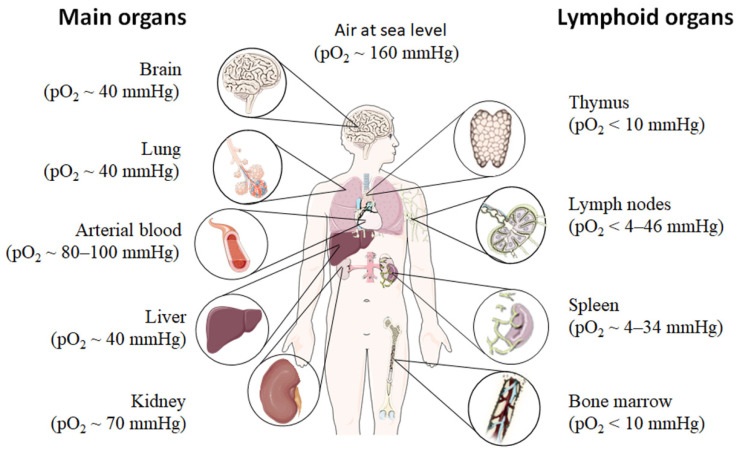
Oxygen partial pressure (pO_2_) varies throughout the human body from main organs to lymphoid tissues. Figure was modified from Servier Medical Art, licensed under a Creative Common Attribution 3.0 Generic License [16].

**Figure 2 biomedicines-09-00260-f002:**
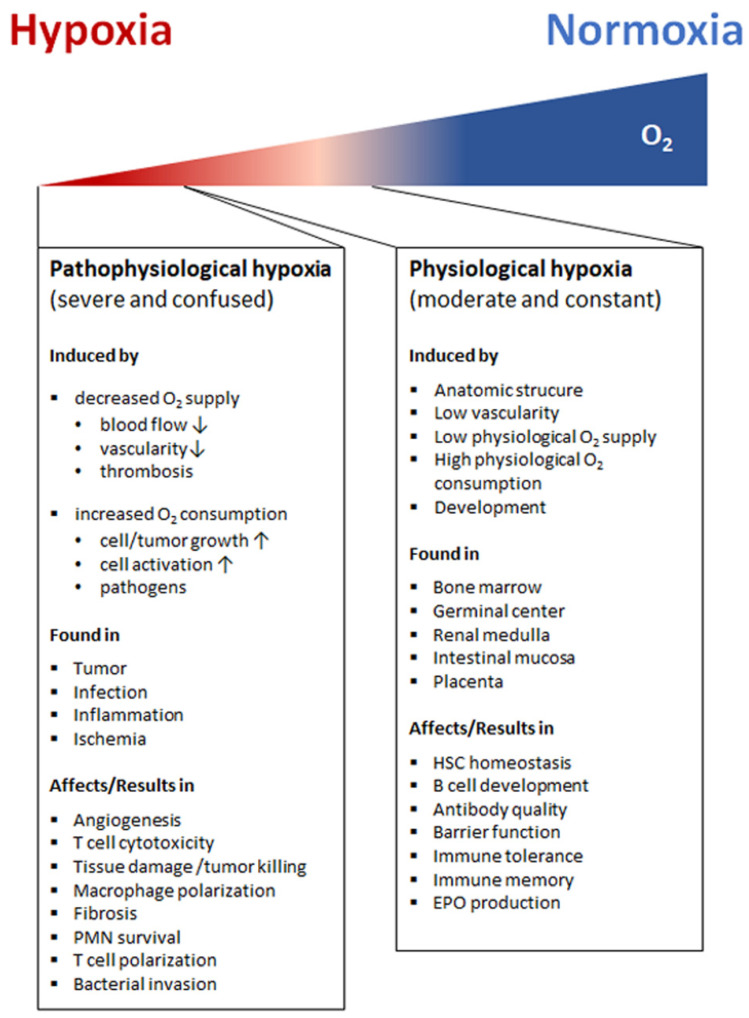
Hypoxia arises in a variety of immunological situations under physiological and pathophysiological immune activity. Physiological hypoxia which is attributed to be moderate and constant is caused by, e.g., oxygen gradients as a result of developmental processes, anatomic structure and functional composition. The factors contributing to physiological hypoxia differ in specific organs such as in primary and secondary lymphoid organs. Here, physiological hypoxia regulates the function of immune cells supporting beneficial immune activity (light red right panel). The pathophysiological hypoxia is usually attributed to be severe and confused, which is caused by a variety of factors in different diseases (dark red left panel). (↑: induced or up-regulated, ↓: reduced or down-regulated. PMN, polymorphonuclear neutrophils; EPO, erythropoietin; HSC, hematopoietic stem cell).

**Figure 3 biomedicines-09-00260-f003:**
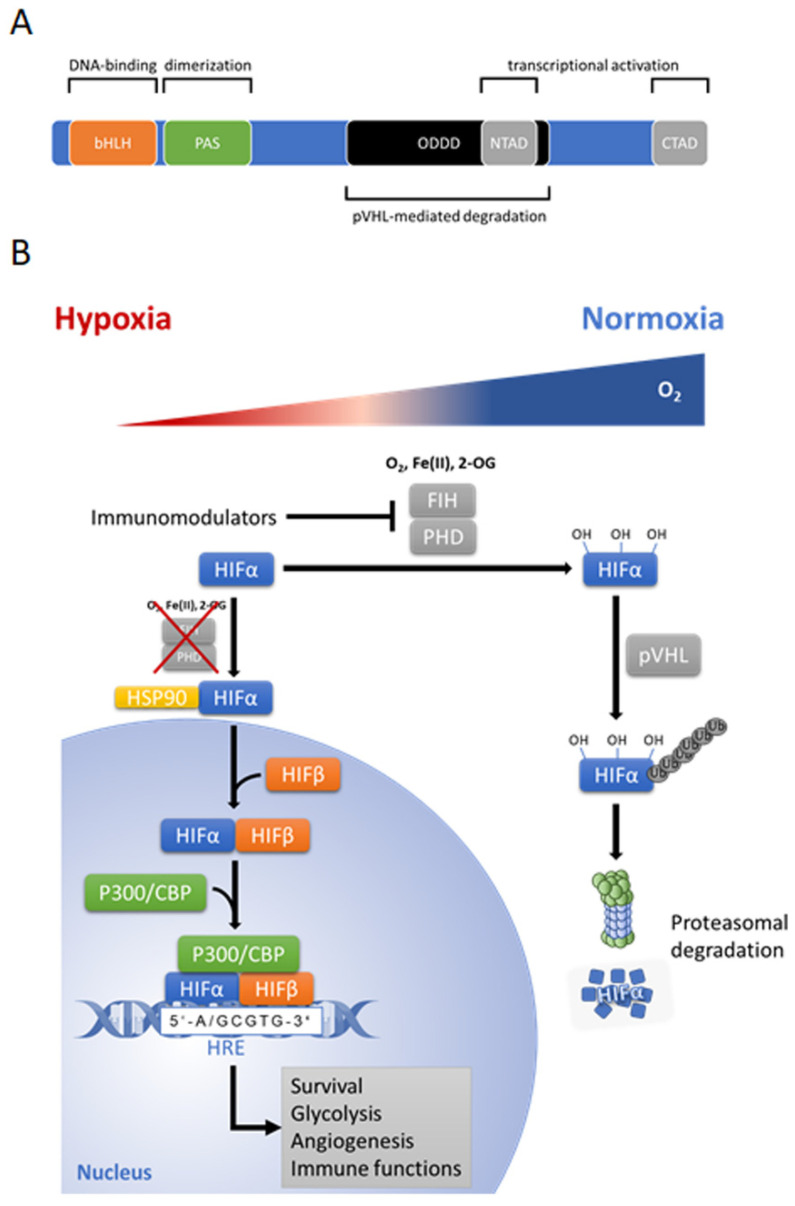
(**A**) Structure of hypoxia-inducible factor (HIF)-1α/HIF-2α subunit. Basic helix-loop-helix (bHLH) domain of HIF-α mediates DNA binding, and PAS domain is required for heterodimerization of the subunits. N-terminal transactivation domain (N-TAD) and the C-terminal transactivation domain (C-TAD) are transactivation domains for activation of HIF target genes. The N-TAD is responsible for the stability and target gene specificity and overlaps with the oxygen-dependent degradation domain (ODDD). The C-TAD interacts with coactivators activating gene transcription. Protein von Hippel–Lindau (pVHL)-mediated degradation of HIF-α needs hydroxylation of two proline residues in (ODDD). (**B**) Cellular HIF-α regulation under hypoxic (red shaded area of triangle) and normoxia conditions (blue shaded area of triangle). In hypoxia, HIF-α subunit translocates into nucleus and dimerizes with constitutively expressed HIF-β subunit, then recruits the transcriptional coactivator p300 and CREB (cAMP response element binding)-binding protein (CBP) to form a complex, which binds to hypoxic response elements (HRE) on DNA and triggers HIF-dependent transcriptional response. Under normoxic condition, when cellular oxygen supply exceeds demand, HIF-α is hydroxylated on prolyl residues by prolyl hydroxylase domain proteins (PHDs), which are members of 2-oxoglutarate (2-OG)-dependent dioxygenase superfamily and function in the present of substrates and co-factors: oxygen (O2), 2-oxoglutarate (2-OG), ferrous iron [Fe (II)] and ascorbate. Hydroxylation facilitates the interaction with the tumor suppressor protein von Hippel–Lindau (pVHL), which recruits an E3-ubiquiine-ligase complex tagging HIF by polyubiquitination for proteasomal degradation via the 26S proteasome. Furthermore, factor-inhibiting HIF (FIH) hydroxylates HIF and inhibits the transcriptional activation of HIF by preventing recruitment of the transcriptional coactivators p300 and CBP. Figure was modified from Servier Medical Art, licensed under a Creative Common Attribution 3.0 Generic License [16].

**Figure 4 biomedicines-09-00260-f004:**
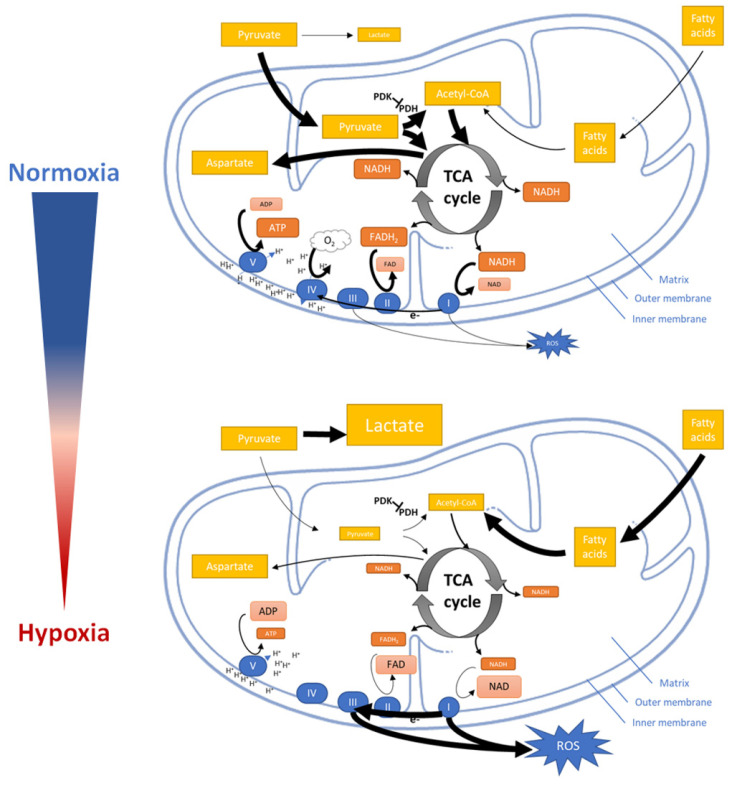
Impact of hypoxia on mitochondrial functions. Under normoxia, pyruvate is shifted into the mitochondria and catalyzed to Acetyl coenzyme A (Acetyl-CoA) as a substrate for the TCA cycle (tricarboxylic acid cycle) to synthesize energy-rich reductants (NADH and FADH_2_) which drive the electron flux through the ETC and provide Aspartate. Under hypoxia, pyruvate is catalyzed to lactate while only few amounts of Acetyl-CoA are provided by fatty acid oxidation to the TCA. Only low amounts of energy-rich reductants and ATP are produced. The lack of the final electron acceptor (oxygen) leads to ROS generation during electron flux through the ETC. (Large arrows indicate higher flux; smaller arrows indicate lower flux; larger icons indicate higher amounts of intermediates; smaller icons indicate lower amounts of intermediates). Figure was modified from Servier Medical Art, licensed under a Creative Common Attribution 3.0 Generic License [16].

**Figure 5 biomedicines-09-00260-f005:**
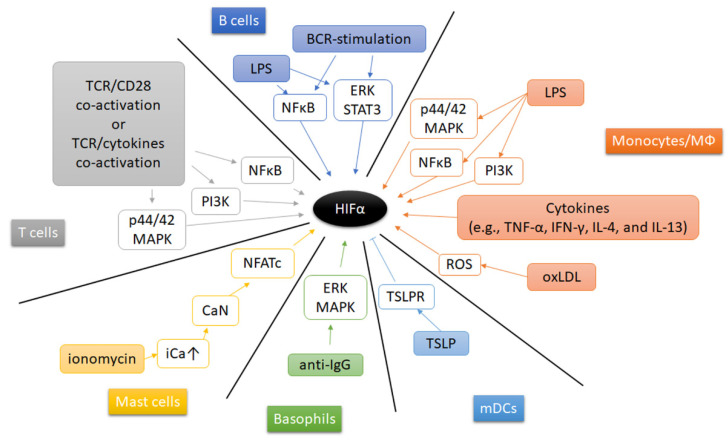
Extracellular regulation of HIF-α. A schematic summarizing the mechanisms underlying the regulation of HIFα activity under diverse physiological and pathological conditions. Arrow indicates activation, and bar-headed line indicates inhibition.

**Table 1 biomedicines-09-00260-t001:** HIF-mediated effects in myeloid cells.

Cell type	HIF Mediated Effects	Ref.
eutrophil granulocytes	**HIF-1:** survival ↑, glycolysis ↑, effector function ↑, (adhesion ↑, migration ↑, production of antimicrobial peptides ↑, formation of DNA traps ↑, phagocytosis ↑, oxidative burst ↑)	[9,80,81,162,163]
	**HIF-2:** survival ↑	[164]
Mast cells, basophils and eosinophils	**HIF-1:** survival ↑, effector function ↑ (synthesis of IL-8, TNF-α, VEGF, and IL-4 following TLR and/or IgE activation), eosinophil chemotaxis ↑	[165,166,167,168,169]
	**HIF-2:** eosinophil chemotaxis ↓	[166]
Monocytes/Macrophages	**HIF-1:** survival ↑, effector function ↑ (release pro-inflammatory TNF-α, IL-1β, IL-12 and IL-6, iNOS activity ↑ and NO production, phagocytosis)↑, glycolysis ↑, M1 polarization	[81,170,171,172]
	**HIF-2:** arginase-1 ↑, M2 polarization, phagocytosis ↓, proinflammatory cytokine/chemokine (IL-1β, IL-12, TNF-α, IL-6, IFN-γ, and CXCL2) expression, migration	[163,170,171,173]
Dendritic cells	**HIF-1:** survival ↑, glycolysis ↑, induction of regulatory T cells ↑, effector function ↑ (migration ↑, proper DC-mediated cytotoxic T cell activation, differentiation ↑)	[174,175,176,177]
	**HIF-2:** unknown	

↑: induced or up-regulated, ↓: reduced or down-regulated.

**Table 3 biomedicines-09-00260-t003:** HIF-mediated effects in innate lymphoid cells.

Cell Type	HIF Mediated Effects	Ref.
Innate lymphoid cells (ILC2)	**HIF-1:** late-stage maturation and function of ILC2s via targeting IL-33-ST2 pathway	[244]
	**HIF-2:** unknown	
NK cells	**HIF-1:** tumor growth ↑ (angiogenesis↑, VEGFR-1↑) tumor cell killing ↑, INF-γ ↓, OCR/ECAR ratio ↓	[245,246]
	**HIF-2:** unknown	

↑: induced or up-regulated, ↓: reduced or down-regulated.

**Table 4 biomedicines-09-00260-t004:** Summary of select ongoing clinical trials of HIF-2α inhibitors in cancer.

Compound/Drug Name	Objective	Type of Cancer	Phase	NCT Number
PT2977 (MK6482) (Inhibition of HIF-2 heterodimerization)	Tumor response	advanced solid tumors/ccRCC /specified solid tumors/glioblastoma (GBM)	I	NCT02974738
	ORR	Advanced or metastatic ccRCC	II	NCT03634540
	ORR	VHL-Associated Renal Cell Carcinoma	II	NCT03401788
PT2385 (Inhibition of HIF-2 heterodimerization)	MTD (Part 1: PT2385; Part 2: PT2385 + nivolumab; Part 3: PT2385 + cabozantinib)	Advanced ccRCC	I	NCT02293980
	Tumor radiographic response	Recurrent GBM	II	NCT03216499
	ORR	VHL disease-associated ccRCC	II	NCT03108066
ARO-HIF2 (Neutralization of *HIF2A* mRNA)	AEs; RP2D	ccRCC	I	NCT04169711

NCT National Clinical Trial (clinicaltrials.gov, accessed on 25 February 2021), RCC, renal cell carcinoma; ccRCC, clear cell renal carcinoma; GBM, glioblastoma; ORR, objective response rate; VHL, Von Hippel-Lindau; MTD, maximum tolerated dose; AEs, adverse events; RP2D, recommended phase 2 dose; PFS, progression-free survival.

## Data Availability

Not applicable.

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
