# Peer review of "Hypoxia/HIF Modulates Immune Responses"

_biomedicines, 2021, doi:10.3390/biomedicines9030260_

Round 1
Reviewer 1 Report
The article described how hypoxia, especially HIFs, modulates immune system in physiological and pathological condytions. The text requires some editing e.g. figures captions are devided on pages.
In paragraph with hypoxia definition the radiobiological definition of hypoxia will be good to mention.
In line 106, I think it will be better to replace "inhaled air" to "surrounding tissues".
Generraly, in the whole text, the cyclic hypoxia was not mentioned. It shoud be good to discus this topic in one paragraf in context of cyclic hypoxia.
Line 198 and 229 - to many space bars.
Fig.4. Plese remove yellow beckground and increase the fonts.
Line 784 - reference error.
Style in Ref. 62, 93, 107 shoud be checked.
Reviewer 2 Report
The authors did an intensive literature review on the selected topic. The text is well written and comprehensive to the readers. The reviewer have few minor suggestions that could improve the manuscript.
- In general the short titles throughout the manuscript doesn’t exactly represents the text inside that. e.g. Line 58, the title says “in lymphoid organs” however, the content also talks about other non-lymphoid organs; line 176, title says “Hypoxia-induced transcriptional changes” whereas the text talks about the structural and functional modifications of HIF. This section also talks about posttranslational regulation and mechanisms including O2-dependent and independent pathways.
- Line 292, section 4.2, the content in this topic is very dense and less relevant, could be shortened or simplified
- Line 348, “Last but not least” this type of languages should be modified
- Line 248-385, though the text is well written about the mitochondrial function and how hypoxia affects its function, this context are less relevant to the selected topic or at least should be placed in a separate sub-title.
- Figure 4, please provide a detailed figure legend as I cannot differentiate a major change between the two mitochondrial images, other than some icons are smaller in size (what does the smaller icon mean?)
- Abbreviations used in the manuscript should be consistent eg, please use either TNF-α or TNF, but not both.
The author used sufficient references.
